# Effects of Two Exercise Programs on Neck Proprioception in Patients with Chronic Neck Pain: A Preliminary Randomized Clinical Trial

**DOI:** 10.3390/medsci11030056

**Published:** 2023-09-08

**Authors:** Leila Rahnama, Manizheh Saberi, Pegah Kashfi, Mahsa Rahnama, Noureddin Karimi, Mark D. Geil

**Affiliations:** 1School of Kinesiology, Nutrition & Food Science, California State University, Los Angeles, CA 90032, USA; 2Department of Physiotherapy, Shiraz University of Medical Sciences, Shiraz 71348-14336, Iran; sabeript@yahoo.com; 3Department of physiotherapy, National University of Medical Sciences, 28001 Madrid, Spain; 4Department of Physiotherapy, University of Social Welfare and Rehabilitation Sciences, Tehran 19857-13871, Irankarimi@uswr.ac.ir (N.K.); 5School of Medicine, Shiraz University of Medical Sciences, Shiraz 71348-14336, Iran; 6Wellstar College of Health Professions and Human Services, Kennesaw State University, Kennesaw, GA 30144, USA; mgeil@kennesaw.edu

**Keywords:** neck pain, proprioception, exercise

## Abstract

Background: The purpose of this study was to compare the effects of specific neck muscle training and general neck-shoulder exercises on neck proprioception, pain, and disability in patients with chronic non-specific neck pain. Methods: Twenty-five patients with chronic non-specific neck pain were recruited into this preliminary single-blinded randomized clinical trial. They were randomly assigned to either a specific neck exercise (n = 13, mean aged 24 years) or a general neck exercise group (n = 12, mean aged 25 years). Specific neck exercises included eye-head coordination and isometric deep neck muscle exercises. General neck exercises included neck and shoulder free range of motion and shoulder shrug. Pain, disability, and neck proprioception, which was determined using the joint repositioning error, were measured at baseline and after eight weeks of training in both groups. Results: Both training groups showed significant improvements in joint repositioning error (*p* < 0.001, F = 24.144, ES = 0.8), pain (*p* < 0.001, F = 61.118, ES = 0.31), and disability (*p* = 0.015, F = 6.937, ES = 0.60). However, the specific neck exercise group showed larger variability in joint repositioning error (*p* = 0.006, F = 0.20, F critical = 0.36). Conclusions: Either specific neck exercise or a general neck-shoulder range of motion exercise could be effective in improving neck proprioception. Therefore, exercises could be recommended based on patient comfort and patients’ specific limitations.

## 1. Introduction

Neck proprioceptors play an essential role in monitoring normal neck and head positions and movements with respect to the trunk. Additionally, they help stabilize head posture via their connections to the vestibular system [1]. Since proprioceptive deficit has frequently been observed following chronic neck pain [2,3], regaining neck proprioception is a critical part of neck pain rehabilitation to decrease the extra reliance and postural dependency on visual and vestibular systems while performing functional tasks.

Several exercise programs have been prescribed for chronic neck pain patients to relieve pain and improve proprioceptive acuity. Arami et al. [4] examined the effects of specific proprioceptive training versus endurance training on pain, muscle strength, and neck proprioception in people with chronic non-specific neck pain (CNNP). They found endurance training to be more efficient in improving neck muscle endurance, while proprioceptive training was more influential in improving neck proprioception. On the contrary, Jull et al. [5] compared two exercise programs (proprioceptive training versus deep neck flexor muscle training) to improve neck proprioception in chronic neck pain, and found that both programs enhanced neck proprioception, except for a minor advantage detected after proprioceptive training. They hypothesized that deep flexor training can improve neck proprioception via reducing pain and strengthening the muscles which stabilize the cervical spine.

Proprioceptive information and inputs are collected by proprioceptive receptors which are located in skin, joint structures, and muscles [6]. Muscle spindles located in muscles are low-threshold receptors which receive proprioceptive information [6]. Deep cervical flexors including longus colli and capitis [7], in addition to deep cervical extensors including multifidus and deep sub occipital muscles, have a large number of muscle spindles and neural connections to the vestibular system and postural reflexes [8]. Therefore, it is believed that these muscles play a major role in providing normal neck proprioception. Alternatively, zygapophysial joint structures, including the joint capsule and surrounding ligaments, have a large number of rapidly adapting mechanoreceptors which provide information about the joint movements and positions [6]. Hence, any exercise that aims to improve neck proprioception should upskill one or more of these proprioceptors.

Despite the effectiveness of specific training in improving neck muscle strength and reducing pain [5,9], many clinicians prescribe general exercises to patients with neck pain, including neck and shoulder range of motion (ROM). Shoulder ROM is prescribed when neck muscle activation was observed during a light shoulder elevation task [10,11]. There are two main reasons for this preference. First, these dynamic general exercises encourage neck movements which help break the pain spasm cycle. Second, the rate of clinician and patient compliance with general exercises is higher compared to specific training programs. Furthermore, most clinicians do not have adequate time to supervise their patients during specific training programs. [10]. 

While regaining neck proprioception is an essential part of neck pain management, prescribing an efficient and simple program is also important. As discussed, neck proprioceptors are located in muscles, joints, and the surrounding fascia [2]. Therefore, we aimed to compare the effects of a specific deep neck muscle training program with a general neck exercise program on neck proprioception, pain, and disability in patients with CNNP. 

## 2. Methods

### 2.1. Participants and Study Design

Twenty-five individuals with CNNP were recruited for this IRB-approved preliminary clinical trial. The recruitment process was performed through spreading IRB approved flyers in the university campus (April 2018–March 2019). This preliminary randomized control trial constitutes an amendment to a previously registered randomized clinical trial, denoted as IRCT2017091620787N2, which was registered in 2017. The protocol for the main study was published in 2019 [12]. After volunteers contacted the main investigator, they were interviewed over the phone for eligibility criteria. Eligible volunteers were randomly assigned to separate exercise groups. Patients with CNNP aged 18 to 50 who reported a pain intensity of greater than 30 mm on visual analogue scale (VAS) and who had not received any exercise therapy in the past six months were included in the study. The age limit of 50 was considered because changes in muscle properties have been observed in older adults [13]. Participants were excluded if they reported a history of cervical spine trauma, whiplash injury, inflammatory diseases, cervical radiculopathy, structural spinal deformity, vertigo, and vestibular disorders [14,15]. Study details, including the main purpose and program procedures, were fully explained to all subjects and written consent was obtained prior to the study procedures. The study flow chart is demonstrated in Figure 1 using CONSORT flow diagram.

### 2.2. Randomization

Upon signing the informed consent, participants were asked to choose a sealed envelope which categorized them to either group A or group B. Group A corresponded to the deep neck muscle exercise group and Group B corresponded to the neck and shoulder general exercise group. Patients in each group were not aware of the other exercise strategy. Randomization was performed by someone who was blinded to the remaining study procedures.

### 2.3. Outcome Measures

Neck proprioception, pain, and disability were the primary outcome measures. Outcome measures were recorded at the first session prior to initiation of the exercise program to establish a baseline, and again after eight weeks of training. Neck proprioception was assessed using joint repositioning error (JRE). To maintain consistency, the same investigator provided exercise training to both groups. For this reason, the investigator used standardized language with both groups, reading from a customized booklet.

Joint repositioning error for neck rotation was measured based on the methods first described by Revel et al. [16]. Participants were invited to sit on a chair which was located 90 cm from a white wall. A laser beam pointer was fastened by an elastic strap to their heads. They were asked to keep their heads in a relaxed and neutral position while they looked straight forward at the wall (Figure 2). The laser light on the wall was marked as the reference point. Then, patients were asked to turn their head to the right and to the left to the end of available range, each one once, and return to the original position with eyes open, attempting to align the laser light with the reference point. Next, their eyes were covered by a blindfold and they were asked to repeat the procedure with their eyes closed. While returning their heads to the original position, participants were asked to inform the examiner when they reached the original position. The new laser point on the wall was marked as the target point. Joint position error was calculated in degrees as the arctangent of the distance between the target and reference point in cm divided by 90 cm. They performed the procedure three times for each right and left side and the mean values of errors for each side were used for further analyses [17,18]. 

Pain intensity was measured in mm using Visual Analogue Scale (VAS) ruler in which 100 is the highest imaginable pain and zero is no pain at all. Patients were asked to indicate their pain intensity on the VAS ruler once at the beginning of the study and once after completing the eight-week exercise program.

The amount of perceived disability by patients was recorded using the Neck Disability Index (NDI) [19]. The NDI, originally developed in 1991, presently stands as the most extensively employed tool for evaluating self-assessed disability among individuals suffering from neck pain. This questionnaire consists of five main areas including pain intensity, ability to complete activities of daily living, and severity of headaches. Within each query, there are six potential responses, ranked on scale of 0 (signifying no disability) to 5 (indicating a complete disability). Cumulative scores are then calculated for all sections. The resulting score is calibrated on a range of 0 to 50, where 0 represents optimal functioning and 50 represents the utmost impairment. Alternatively, this score can be presented within a 0 to 100 range, frequently denoted as a percentage (0–100%).

For employing the NDI in clinical decision making, a change of 5 points was established as clinically significant, displaying a sensitivity of 0.78 and specificity of 0.80.

### 2.4. Intervention

The intervention included two exercise programs. Both were prescribed over three sets daily, three days a week, for eight consecutive weeks [20]. One daily exercise set was supervised by a trained physical therapist and the other two sets were performed at home without any external supervision. Each set started at five repetitions of exercises and increased to 20 repetitions by week eight. No specific order for performing exercises was required. An exercise booklet that included all exercise program details was given to each participant during the first session.

#### 2.4.1. General Neck Exercise Program

General neck exercises (GNE) included neck active range of motion (ROM) in all directions consisting of flexion, extension, right and left rotation, and right and left side bending. Shoulder active ROM included shoulder flexion and abduction up to 90° and external rotation/internal rotation while the elbow was 90° flexed and the arms were beside the body. All participants were advised to complete their active ROM exercises in the sitting position without holding at the end range. They also performed a shoulder shrug exercise while they were standing. In addition, participants were asked to sit on a chair and put their hands behind their legs, and then lift their bodies with their hands. All exercise/movement directions were repeated five times during the first session and gradually increased to 20 repetitions through the eight-week program [9]. Participants were instructed to not hold at the end of active ROM exercises. The decision to abstain from instructing participants to sustain the end of the ROM was made in order to prevent triggering any stretch reflexes and to avoid the potential induction of pain. Instead, participants were directed to perform the exercise encompassing the complete ROM available in each direction.

#### 2.4.2. Specific Neck Exercise Program

Specific neck exercises (SNE) including deep neck muscle strengthening were performed in a crook lying position with no specific order. Patients were instructed to (a) move their eyes upward and backward without any head and neck movements and hold for 5 s at the end range, (b) move their eyes forward and downward without any head and neck movements and hold for 5 s at the end range, (c) do a very mild chin tuck with five seconds hold at the end range, (d) do a mild resistive chin tuck with a mild resistance to their chins by their hands and hold for 5 s at the end range, and (e) press their occiput to the bed with submaximal force and hold it for 5 s. Each exercise was initially performed 5 times and increased to 20 times by the end of the eight-week program [5,9].

### 2.5. Statistical Analysis

Based on the previous research on neck proprioception in patients with neck pain [21] and the significant difference for joint repositioning error (JRE) = 1.8° and the SD = 1.64°, α = 0.05, and β = 0.2, the sample size using the below formula was estimated at N = 26.
2σ2×(1.96+0.84)2D2

Data analyses were performed using SPSS (v. 26, IBM, Armonk, NY, USA) and Excel 2016 (Microsoft, Redmond, WA, USA). The Shapiro–Wilk test was used to test the normality assumption. Mixed Factorial ANOVA was used to evaluate the effects of the interventions on outcome measures through time and between the groups. Within-subject factors were side (right and left) and time (before and after). The between-subject factor was group (GNE and SNE). F-tests were used to compare the variability of JRE between groups. Mauchly test was conducted using SPSS to assess the violation of sphericity and Cohen’s d was calculated to estimate effect size. Effect size was classified as small (d = 0.2), medium (d = 0.5), and large (d > 0.8). The level of significance was set at 0.05. 

## 3. Results

Thirteen subjects with an average age of 24 ± 4 years were included in the SNE group and twelve subjects with an average age of 25 ± 3 years were assigned to the GNE group. Participants age range were 20 to 28 years old and were all female. 

The Shapiro–Wilk analysis revealed a normal distribution of all variables except for pain after the specific training program. However, it was approximately normally distributed. Therefore, the Mixed ANOVA assumption was not violated. 

Pain, Disability, and Proprioception

Pain decreased significantly in both groups (*p* < 0.001, F = 61.118, ES = 0.31) with 95% CI = 1.95–3.52 for the estimated marginal mean differences but there was no difference between groups (*p* = 0.682). In the GNE group, 41.5% of participants reported a fifty percent reduction in pain, while the proportion of SNE participants was 38.5%. 

The difference between groups was more pronounced in self-reported NDI disability results (Figure 3). At baseline, the means for the groups were practically identical (33.17 vs. 33.28). Both groups significantly improved over time (*p* = 0.015, F = 6.937, ES = 0.60) with 95% CI = 1.66–13.81 for the estimated marginal mean differences. However, the improvement for the SNE group was only 2.3, compared to an improvement of 13.17 for the GNE group. Nonetheless, this group difference was not statistically significant (*p* = 0.07, F = 3.416).

Both exercise programs resulted in improvements in all outcomes. No significant interaction was seen. There were no significant differences between the left and right directions for JRE, so the mean of the two directions was used for further analysis. The average repositioning error was reduced by more than 1.2° in both groups (*p* < 0.001, F = 24.14, ES = 0.8) with 95% CI = 1°–2.45° for the estimated marginal mean differences (Table 1). However, there was no significant difference between groups (*p* = 0.585). Notably, the SNE group demonstrated greater variability in comparison to the GNE group (*p* = 0.006, F = 0.20, F critical = 0.36) (Figure 4). 

## 4. Discussion

This study investigated the effects of deep neck muscle training compared to the general range of motion exercises on pain, disability, and proprioception acuity in patients with neck pain. We found that both exercise programs not only improved pain and disability but also increased proprioception acuity. Neck proprioception plays a significant role in maintaining balance, coordinating movements, and establishing precise body diagram. 

### 4.1. Pain

Our results are in agreement with the previous literature in which deep neck muscle training [8] or functional training [22] improved proprioception and reduced pain and disability in patients with neck pain. These changes could occur due to metabolic adjustments in neck muscle cells after exercise therapy. Patients with chronic neck pain experience mitochondrial damage, which in turn causes reduced concentration of the Na^+^-K^+^ pump in neck muscles, including the trapezius. In addition, a reduced blood flow has been observed in this muscle in chronic neck pain [23,24]. These alterations in the Na^+^-K^+^ pump concentration and blood flow could be the reason for feelings of fatigue in CNNP patients, which in turn results in disability [23]. However, exercise increases the concentration of the Na^+^-K^+^ pump and blood flow in muscles, which might contribute to the reduced pain and disability observed after exercise therapy [23,24]. Furthermore, exercise leads to increased metabolism in muscle cells which can modify the muscle tissue environment, in which nociceptor activity diminishes and leads to decreased pain sensitivity. Moreover, muscle contraction can activate the stretch reflex receptors and cause an increase in endorphin release from the central nervous system [25]. The increase in endorphin level reduces the perceived pain and disability in sequence. Considering the minimal clinically important difference (MCID) of 9.9 (10) mm for pain on VAS [26], both exercise groups showed clinically significant effects for pain reduction (>29 mm). 

### 4.2. Disability

We observed a tendency toward a more remarkable reduction in disability in the GNE group. This finding can be explained by the modification of catabolic processes brought by mobility. Immobility induces decreased protein production and increased catabolic procedures which might, consequently, irritate nociceptors [24]. The neck and shoulder ROM exercises in our GNE program increased the neck-shoulder mobility and probably led to increased protein turnover in the trapezius muscle and decreased the disability in this group by a greater amount. It is even more remarkable when we compare the MCID for NDI, ranging from 13 to 18 points (28), with the before-after intervention NDI changes in the present study. We observed an NDI difference of 13.1 in the GNE group versus 2.3 in the SNE group. These results are in line with the findings of Rolving et al. [27], who observed improvement in pain and disability in both a general physical activity group and a neck and shoulder specific training group in patients with neck pain. Additionally, these findings agree with those of Dedering et al. [28], who found both prescribed physical activity and neck specific training effective in reducing pain and disability in patients with cervical radiculopathy. Beyond the modification of the catabolic processes prompted by exercise, any articulation motion or muscular activity amplifies the blood circulation in the region. This, in turn, facilitates the removal of catabolic byproducts and promotes the influx of oxygen-rich blood, ultimately ending in reduced nociceptor activity, diminished pain, and improved precision in movements.

### 4.3. Proprioception

Aside from pain and disability reduction, we observed proprioception improvement in both training programs. These findings emphasize the findings by Mahto and Malla [29], who found both deep neck flexor strengthening and proprioceptive neuromuscular facilitation exercises effective in reducing neck JRE in patients with neck pain. Additionally, in agreement with our findings, Humphreys et al. [30] suggested that a simple eye-head-neck coordination exercise may help improve neck proprioception. 

Proprioceptive deficits observed in CNNP patients can impair the eccentric activity of neck muscles, which in turn leads to reduced neck stability and excessive strain and microtrauma to the neck structures. Muscle contraction, in contrast, can modify the activity of Golgi tendon organs, their adaptation to stretch, and α-γ coactivation. All these result in better sensorimotor control, movement patterns, and proprioceptive acuity after exercise [24]. In addition, muscle contraction enhances metabolic behavior in intrafusal muscle fibers, including muscle spindles, which are partly responsible for proprioceptive accuracy [31]. Mild resistance training also increases the sensitivity of muscle spindles, tendon organs, and joint proprioceptors [31]. Therefore, stimulating these mechanoreceptors leads to an increase in afferent nerve activity while inhibiting the pain nociceptors, according to the gate theory. 

Despite all the benefits of muscle spindle retraining on proprioception acuity, we observed more variation in JRE improvement in the SNE group. This finding may be interpreted as lesser stability in JRE improvement after SNE training. Three of the patients in the SNE group even showed increased JRE after the training, which was unexpected. The reason for this finding could be the number and type of proprioceptors which are upskilled in each training program. Eye-head coordination exercises such as the exercises in our SNE groups can improve proprioceptive acuity by stimulating the muscle spindles; in deep neck muscles, the spindles have several connections to the vestibular and visual systems. On the other hand, ROM exercises such as those in the GNE group can stimulate Golgi tendon organs and muscle spindles as well as joint mechanoreceptors. Therefore, it seems that the GNE program retrained more proprioceptors and caused a more stable improvement in the proprioception acuity compared to the SNE program. Some researchers believe that CNNP patients have a variety of sensorimotor impairments and a single mode of exercise cannot address all of them [32]. Accordingly, it seems reasonable to have a more stable improvement in proprioception by GNE exercises which target different body regions, muscles, and proprioceptors. 

Furthermore, there are various methods for quantifying proprioception. Within the scope of this investigation, JRE was utilized to indirectly evaluate the precision of joint repositioning. Given the role of joint receptors in furnishing this kind of sensory input, it is conceivable that incorporation of the exercises involving joint motions (GNE) could have had more consistency in diminishing the JRE.

The intrinsic muscles of the eye and neck are enriched with muscle spindles, underscoring the significance of accurate ocular movements and the ongoing necessity for precise head positioning. As a result, the SNE program targeting both the intrinsic eye and neck muscles contributed to an enhancement of proprioception as well.

Our findings could align with the suggested approach for addressing chronic neck pain, as outlined in the clinical practice guidelines for neck pain. These guidelines advocate for the use of diverse and combined exercise regimens instead of relying on a solitary exercise. A mixed exercise program for the cervical and scapulothoracic regions may encompass neuromuscular exercises (such as coordination, proprioception, and postural training), stretching, strength training, endurance exercises, aerobic conditioning, as well as elements that engage cognitive and emotional aspects [33].

## 5. Limitations

Our participants showed a disability index score of about 33% at baseline, which is considered a minimal disability. We are not sure if patients with a higher level of disability can follow and perform our exercise programs and obtain the same results. In addition, our sample of CNNP patients were young and active, with a mean age range of 20–28. Therefore, older adults may not benefit from these training programs in the same manner. Furthermore, it remains uncertain whether these changes endure over an extended duration or the effects are more likely to be of a medium-term nature.

The data we have obtained originate from a cohort of 25 individuals, 12 participants in one group and 13 participants in the other. To enhance the accuracy of our findings, it is advisable to work with a larger sample size. Additionally, incorporating a 6 month follow-up period would allow assessing the durability of the observed outcomes.

## 6. Conclusions

Our findings show that both general neck-shoulder ROM and deep neck muscle exercises including eye-head coordination exercise can potentially be effective in improving neck proprioceptive acuity, pain, and disability in CNNP patients. Accordingly, any of these exercise programs do not have superiority in improving neck proprioception acuity as we hypothesized, and exercise might be prescribed based on patient preferences of comfort, ease, and limitations. It is important to mention that certain patients find it challenging to engage specific exercises as they require focus and deliberate performance, leading them to opt for more general exercise routines. Conversely, some patients may experience increased pain from excessive neck and shoulder movements, making them lean towards a specific exercise program. Our findings underscore the significance of tailoring decisions to each individual’s needs.

Nevertheless, as previously mentioned, the evaluations were conducted after an 8 week exercise period without a subsequent long term follow up. This prevents us from making a conclusion on the impact of either exercise regimen or its potential for sustained effects over time.

## Figures and Tables

**Figure 1 medsci-11-00056-f001:**
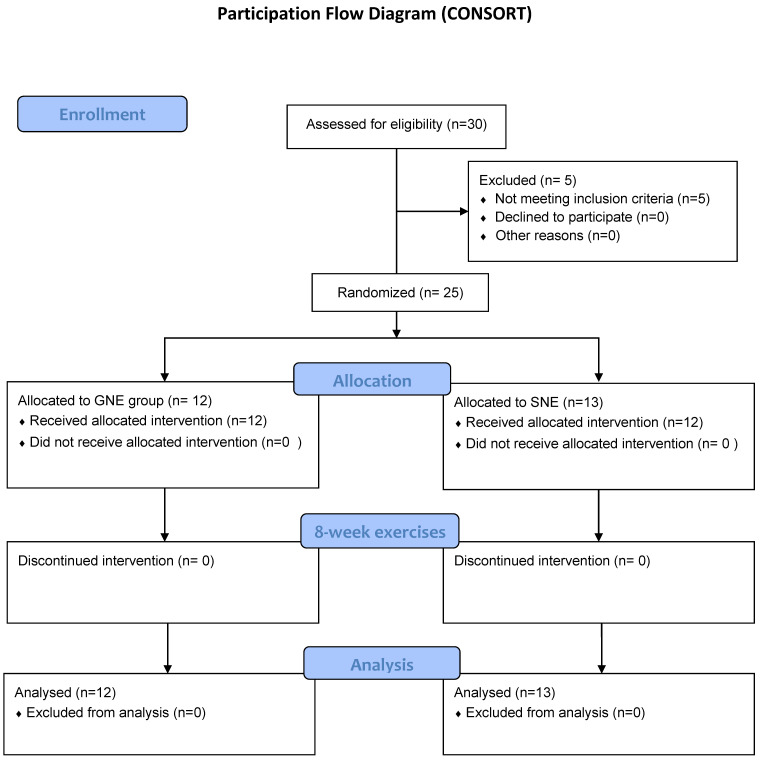
CONSORT Flowchart.

**Figure 2 medsci-11-00056-f002:**
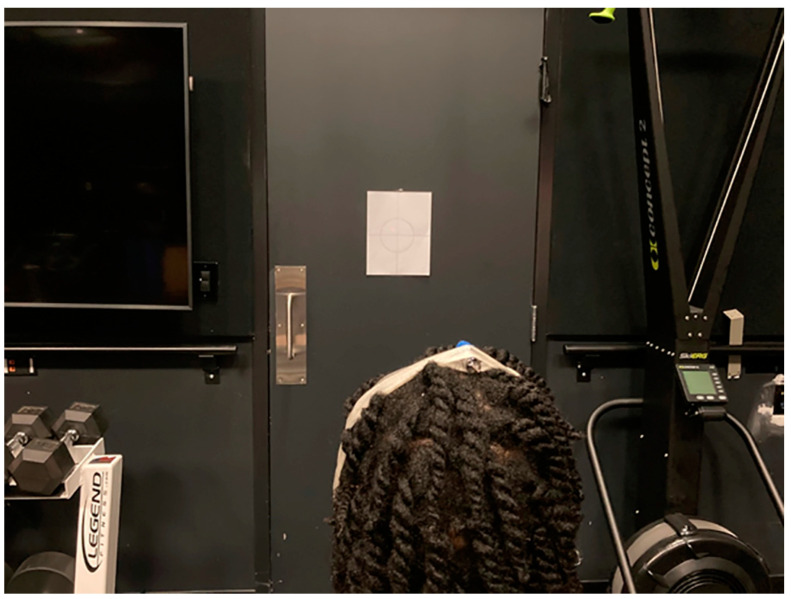
Neck proprioception (repositioning error test).

**Figure 3 medsci-11-00056-f003:**
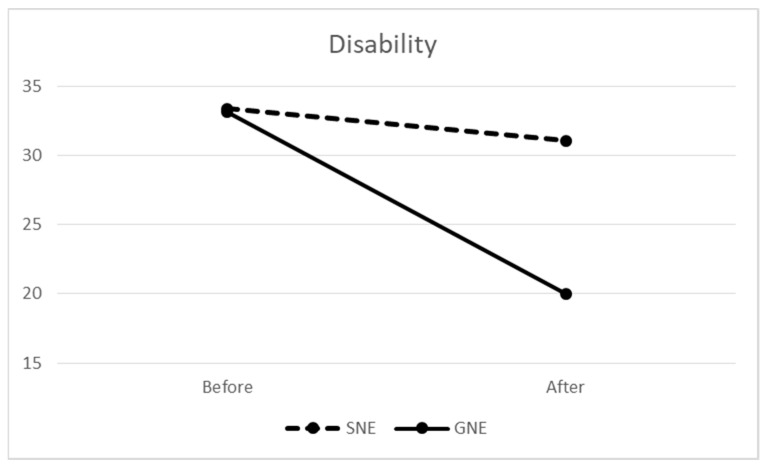
Neck disability changes after neck exercise in specific neck exercise (SNE) and general neck exercise (GNE) groups.

**Figure 4 medsci-11-00056-f004:**
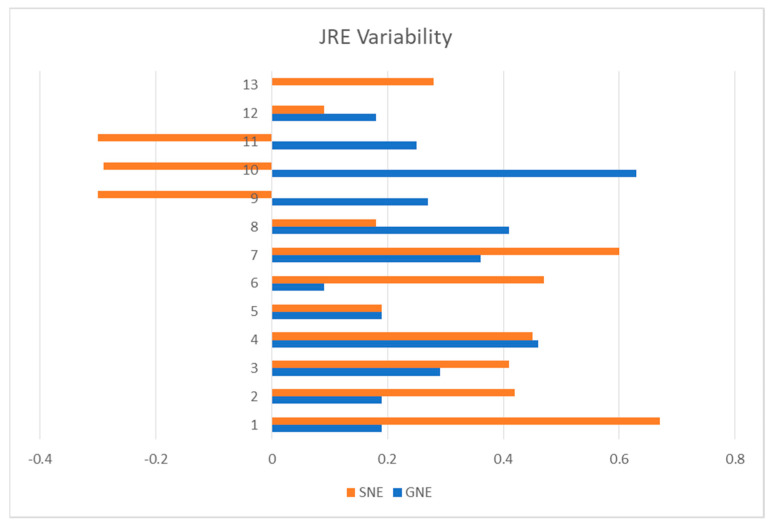
Joint repositioning error (JRE) variability in specific neck exercise (SNE) and general neck exercise (GNE) groups.

**Table 1 medsci-11-00056-t001:** Mean and standard deviation of the neck joint repositioning error (JRE) in both groups.

Time	Outcome	GRE (n = 12)	SNE (n = 13)
	JRE left (degree)	5.62 ± 2.91	4.47 ± 2.62
	JRE right (degree)	5.66 ± 1.67	5.65 ± 1.67
	Pain (mm)	60.75 ± 10.29	60.23 ± 10.83
Before	Disability	33.17 ± 16.17	33.38 ± 20.95
	JRE left (degree)	3.97 ± 2.47	3.49 ± 2.44
	JRE right (degree)	3.72 ± 1.08	3.67 ± 1.95
	Pain (mm)	30.83 ± 10.47	30.85 ± 20.19
After	Disability	20.00 ± 13.32	31.08 ± 19.64

## Data Availability

The data presented in this study are available on request from the corresponding author. The data are not publicly available due to privacy.

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
