# Peer review of "Effects of Two Exercise Programs on Neck Proprioception in Patients with Chronic Neck Pain: A Preliminary Randomized Clinical Trial"

_medsci, 2023, doi:10.3390/medsci11030056_

Round 1

Reviewer 1 Report

The article Title"Effects of two exercise programs on neck proprioception in patients with chronic neck pain: A preliminary randomized clinical tria.  The purpose of this study was to compare the effects of specific neck muscle training and general neck-shoulder exercises on neck proprioception, pain, and disability in patients with chronic non-specific neck pain. The conclusions either specific neck exercise or general neck-shoulder range of motion exercise could be effective in improving neck proprioception. The main question addressed by the research is clear. The topic is original  and it add new information. It seems that no specific improvements should the authors consider regarding the methodology. The conclusions consistent with the evidence and arguments and  they address the main question. The references are appropriate. The tables and figures clear ilustrated the topic.  

Author Response

Response to the Reviewers:

We are grateful for the valuable feedback provided by the reviewers. In response, we have made appropriate revisions to the manuscript. The specifics of these updates are outlined below for your reference.

Reviewer 1:

Comment: It seems that no specific improvements should the authors consider

Response: Thank you for your valuable comments on the previous revisions that helped the article improve to this level.

Reviewer 2 Report

Dear authors,

I appreciate the changes in the manuscript based on the comments. There are still some improvements:

-The flowchart must be completed.

-The conclusions have to respond to the aim of the study and present a more specific comparison that helps therapist to choose each option based on patient's peculiarities rather than his/her preferencies.

No issues detected regarding English language.

Author Response

Response to the Reviewers:

We are grateful for the valuable feedback provided by the reviewers. In response, we have made appropriate revisions to the manuscript. The specifics of these updates are outlined below for your reference.

Reviewer 2:

Comment: The flowchart must be completed.

Response: We appreciate your attention to this matter. The flow chart has already been finalized and is visible in the Word document. However, there seems to be an issue, possibly IT-related, preventing its visibility in the PDF version.

Comment: The conclusions have to respond to the aim of the study and present a more specific comparison that helps therapist to choose each option based on patient's peculiarities rather than his/her preferencies.

Response: We appreciate the comment. We updated the conclusion accordingly on page 10, lines 346 – 351, highlighted in green:

It is important to mention that certain patients find it challenging to engage specific exercises as they require focus and deliberate performance, leading them to opt for more general exercise routines. Conversely, some patients may experience increased pain from excessive neck and shoulder movements, making them lean towards the specific exercise program. Our findings underscore the significance of tailoring decisions to each individual’s needs.
